# Exploring Zero-Shot Emergent Communication in Embodied Multi-Agent Populations

## Abstract

Effective communication is an important skill for enabling information exchange and cooperation in multi-agent settings. Indeed, *emergent communication* is now a vibrant field of research, with common settings involving discrete cheap-talk channels. One limitation of this setting is that it does not allow for the emergent protocols to generalize beyond the training partners. Furthermore, so far emergent communication has primarily focused on the use of symbolic channels. In this work, we extend this line of work to a new modality, by studying agents that learn to communicate via actuating their joints in a 3D environment. We show that under realistic assumptions, a non-uniform distribution of intents and a common-knowledge energy cost, these agents can find protocols that generalize to novel partners. We also explore and analyze specific difficulties associated with finding these solutions in practice. Finally, we propose and evaluate initial training improvements to address these challenges, involving both specific training curricula and providing the latent feature that can be coordinated on during training.

## 1 Introduction

The ability to communicate effectively with other agents is part of a necessary skill repertoire of intelligent agents and, by definition, can only be studied in multi-agent contexts. Over the last few years, a number of papers have studied *emergent communication* in multi-agent settings (Lazaridou et al., 2016; Havrylov & Titov, 2017; Cao et al., 2018; Bouchacourt & Baroni, 2018; Eccles et al., 2019; Graesser et al., 2019; Chaabouni et al., 2019; Lowe et al., 2019b). This work typically assumes a symbolic (discrete) cheap-talk channel, through which agents can send messages that have no impact on the reward function or transition dynamics. A common task is the so called *referential game*, in which a *sender* observes an *intent* needing to be communicated to a *listener* via a *message*.

In these cheap-talk settings, the solution space typically contains many equivalent but mutually incompatible (self-play) policies. For example, permuting bits in the channel and adapting the receiver policy accordingly would preserve payouts, but differently permuted senders and receivers are mutually incompatible. This makes it difficult for independently trained agents to utilize the cheap-talk channel at test time, a setting which is formalized as zero-shot (ZS) coordination (Hu et al., 2020).

In contrast, we study how gesture-based communication can emerge under realistic assumptions. Specifically, this work considers emergent communication in the context of embodied agents that learn to communicate through actuating and observing their joints in simulated physical environments. In other words, our setup is a referential game, where each message is a multi-step process that produces an entire trajectory of limb motion (continuous actions) in a simulated 3D world.

Not only does body language play a crucial role in social interactions, but furthermore, zoomorphic agents, robotic manipulators, and prelingual infants are generally not expected to use symbolic language to communicate at all. From a practical point of view, it is clear that our future AI agents will need to signal and interpret the body language of other (human) agents, *e.g.*, when self-driving cars decide whether it is safe to cross an intersection.

With that, there has been work on the emergence of grounded language for robots (Steels et al., 2012; Spranger, 2016). To the best of our knowledge however, we are first to explore deep reinforcement learning for emergent communication in the context of embodied agents using articulated motion.

Moreover, while cheap-talk is a great proxy for symbolic communication across dedicated channels, communication through articulated motion means agents have to control their joints to signal. One universal feature of this physical actuation is it requires the expenditure of energy, which is a scarce resource both for biological agents and man-made robots.

Another ubiquitous factor of the physical world (and many other domains) is that communicative intents are *not* distributed uniformly. In particular, the Zipf distribution (Zipf, 2016) is known to be a good proxy for many real-world distributions associated with human activity.

A consequence of combining energy cost with a non-uniform distribution over intents in the

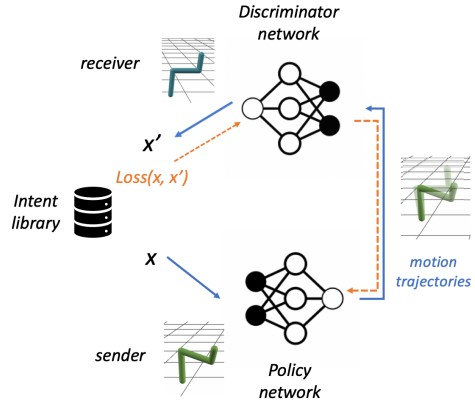

Figure 1: Embodied Referential Game

context of referential games is that, in principle, it allows for ZS communication: Trajectories requiring lower energy exertion should be used for encoding more common intents, while those associated with higher energy encode less common ones.

In contrast, superficially related, auxiliary losses such as entropy penalties do not allow for ZS coordination without further assumptions. While these auxillary losses are design decisions, energy cost is an example of a universal (common-knowledge) cost grounded in the environment, which can be exploited for ZS communication.

Unfortunately, training agents that can successfully learn these strategies is a difficult problem for current state-of-the-art machine learning models. There are three major challenges: 1) Local optima associated with the lock-in between sender and receiver: The interpretation of a message depends on the entire policy, not just the state and action. While recent methods have been developed to address this in discrete action spaces (Foerster et al., 2019), to the best of our knowledge none have been proposed for costly, continuous action spaces. 2) The latent structure underlying the protocol, in our case energy, that can be coordinated on, needs to be discovered, requiring agents to ignore other (redundant) degrees of freedom. 3) Even when this structure is provided, optimization is difficult since it contains a large number of local optima. As a consequence, on top of the continuous optimization problem there is a combinatorial problem of ordering the energy values for each intent.

We explore and analyse these difficulties, suggesting initial steps for addressing them in our setting. First, we show that providing the latent variable (energy) at training time does indeed allow for some amount of ZS coordination. To do so, we adapt our method two fold: 1) During training, we change the observation to only the energy value of a given trajectory. 2) We add an *observer agent* that trains on an entire population of fully trained Self-Play (SP) agents. To evaluate ZS performance, we test this observer on an independently trained set of SP agents. The ZS performance in this setting is around 35% for 10 intents, comparable with the most frequent class for the Zipf-distribution (34%).

Next, we pretrain sender agents to minimize the energy associated with each intent, but ZS performance remains around 34% (10 intents). This is intuitive due to the challenges associated with re-ordering energy values on a 1D line without incurring a high cross-entropy while the different distributions are overlapping. Finally, we show that using the entire trajectory during SP, but only the energy value for the external observer, in combination with pretraining, consistently achieves a much higher ZS performance of around 56%.

All of this illustrates that learning an optimal ZS policy given a specific set of assumptions about the problem setting, which are common knowledge to all parties, is a challenging problem, even in seemingly simple instances. Furthermore, since this work focuses on the two extreme ends of this problem (self-play and ZS), our ideas are relevant for a broad range of intermediate settings as well.

## 2 BACKGROUND

### 2.1 MULTI-AGENT REINFORCEMENT LEARNING

We formalize the protocol learning problem as a decentralized partially observable Markov decision process with $N$ agents (Bernstein et al., 2002), defined by tuple $(S, A, T, R, O, \Omega, \gamma)$. $S$ is the set of states, $A_1 \cdots A_N$ the set of actions for each of $N$ agents in the population, and $T$ a transition function $S \times A_1 \cdots A_N \rightarrow S$ mapping each state and set of agent actions taken to a distribution over next states. This work assumes all agents have identical state and action spaces. In a partially observable setting, no agent can directly observe the underlying state $s$, but each receives a private observation $o_i \in \Omega$ correlated with the state. $O(o|a, s')$ is the probability that $o$ is observed, given action $a$ led to state $s'$. Agent reward $r_i : \Omega \times A_1 \cdots A_N \rightarrow \mathbb{R}$ is a function of state and actions taken. The objective is to infer a set of agent policies that maximize expected shared return $R$. We employ a centralized training regime with execution decentralized (Foerster et al., 2016; Lanctot et al., 2017; Rashid et al., 2018).

Policy gradient algorithms (Williams, 1992; Sutton et al., 2000) are widely used for reinforcement learning domains with continuous action spaces. Similar to (Mordatch & Abbeel, 2018), we use a model-based policy gradient approach with a fully differentiable dynamics model for our emergent communication framework. In particular, we employ a stochastic value gradient approach, SVG-infinity (Heess et al., 2015). SVG methods compute the policy gradient through backpropagation.

### 2.2 ZERO-SHOT COORDINATION

Zero-shot (ZS) coordination is the problem setting of agents coordinating at test time with novel partners, *i.e.* other independently trained agents (Hu et al., 2020). In cooperative multiagent settings, a key challenge is for agents to learn *general* skills for coordinating and communicating with other agents. Nonetheless, just as in single-agent settings, agents can overfit to their environment, in multiagent settings, agents can co-adapt with and overfit to their training partners. Thus ZS coordination is useful for evaluating how well the learned agent policies generalize to unseen agents they may need to later coordinate with (*e.g.* new human partners).

## 3 PROBLEM SETTING

This work is framed in the *ZS Communication* problem setting. ZS communication is a specific instance of the ZS coordination setting, where the task for the agents is to communicate effectively; there is no additional cooperative task being solved concurrently, in the environment. The protocol learning problem then for all independently trained sender-receiver agent pairs is to converge upon optimal zero-shot policies, for communicating intent to novel partners at test time. Sender-receiver pairs engage in *self-play* to train communication protocols, then we use *cross-play* as our evaluation metric for testing ZS coordination. Cross-play (CP) is an instance of out-of-distribution testing, whereby senders are paired with receivers they have never encountered nor influenced during training (*i.e.* agent pairs tested in CP have been trained independently *of one another*).

In our *embodied* problem formulation, populations are composed of spatially articulated agents. Concepts represent communicative intents and messages are instantiated as motion trajectories. Intents are represented as a discrete-valued symbols, defined a priori from an intent library. The policy state space $S$ consists of the joint configuration of an agent and an intent to be communicated, where the joint configuration of the agent is defined by three-dimensional position $(p_x, p_y, p_z)$ and three-dimensional rotation $(r_x, r_y, r_z)$ of each agent joint $j$ in the agent's set of joints $J$. The policy action space $A$ consists of the angular velocities for each joint $j \in J$: $(\Delta r_x, \Delta r_y, \Delta r_z)$. All $a \in A$ are communication actions which can be observed by other agents but have no effect on the environment.

We employ referential games for generating communicative motion through paired-play. At each time step $t$, the policy inputs the joint configuration of the agent, $(p_x, p_y, p_z, r_x, r_y, r_z) \; \forall j \in J$, concatenated with a fixed intent embedding. It outputs an action $a_t$, $(\Delta r_x, \Delta r_y, \Delta r_z) \; \forall j \in J$. Transitions $T : (s_t, a_t) \rightarrow s_{t+1}$ are deterministically computed through a differentiable forward kinematics (FK) module, ensuring kinematically valid trajectories are generated. After $T$ steps, the episode terminates, and the sequence of joint states visited and velocity actions taken is concatenated to compose a motion trajectory. The observer model takes the motion trajectory as input and predicts the actor's intent. The shared return for the episode is the cross-entropy loss between the observer's prediction

and the ground truth intent; this return $R$ is backpropagated from observer to actor, since we use a centralized training regime for the multi-agent system. The FK module has been adapted from an existing motion library (Holden et al., 2017) to be differentiable. Figure 1 illustrates the *embodied* referential game. Algorithm 1 provides an overview of training for the game.

Importantly, for continuous domains, the injection of noise is critical. During trajectory generation, Gaussian noise is added to each action degree of freedom, both to represent imperfect observability and to induce exploration. Noise helps regularize the communication channel, in that a finite noise level essentially provides a quantization of the continuous space, a 'coarse graining', such that trajectories mapped to different intents are distinguishable. This prevents a collapse of the trajectories.

## 4 METHODOLOGY

Given that we aim to develop agents with *general* communication skills (policies that are *not* simply locally optimal to their current communication partner), it is useful to first consider why converging on an embodied protocol that generalizes to novel partners is a challenging problem.

Since embodied communication is *physically* instantiated, it occurs through manipulating the velocities of agent joints; thus, communication through motion generation implies a *high-dimensional continuous* action space. Combining the training criterion with such a high dimensional action space results in a highly non-convex optimization surface. This means the optimization landscape has many local optima (reasonable solutions) for inferring a protocol. It can be difficult for a local optimization algorithm to navigate this type of landscape in search of a *global* optimum. It would also be sensitive to where in the policy space policy parameters are initialized. Because of these challenges, our approach proposes to induce latent structure during protocol training, to provide similar grounding for how independent actors generate communication. This structure can subsequently be exploited to improve ZS coordination at test time.

### 4.1 INDUCING IMPLICIT LATENT STRUCTURE THROUGH PHYSICAL ENERGY EXERTION

To implicitly induce latent structure in the policy learning process (and thus trajectory generation), we propose to use Energy Regularization coupled with a Zipf distribution over intents. The regularizer enforces minimal energy trajectories for the protocol, and Zipf imposes a monotonic ordering upon intents, based upon likelihood. Coupled, they incentivize an inverse relationship between energy exertion and intent frequency, assigning minimum energy trajectories to maximally occurring intents. Moreover in principle, there exists a global optimum for the protocol: energy values can be ordered to be strictly increasing, and intents can be ordered to be strictly decreasing. Then presumably, a 1:1 mapping between energy values and intents can be induced.

In practice however, this is a very challenging optimization problem, primarily because the optimization surface has many local optima for inferring a protocol. Given the latent structure, there is additionally the problem of strictly ordering energy values, which is combinatorial in the number of intents. Different local optima will solve this problem in different ways, and because energy values are continuous, reasonable solutions can be obtained (and achieve low loss) without matching the correct ordering. This is especially true as the number of concepts to be learned grows and increasing numbers of intents have very similar likelihoods of occurring. Additionally, because the learning problem for each agent is to infer *one* policy that effectively communicates *multiple* intents, if the optimization algorithm gets stuck in a local optimum, finding better local optima likely requires non-local traversal through the policy space. This is because the algorithm may need to change the actions for multiple intents concurrently, to locate a better solution in the space.

For inducing implicit structure, there are two objectives traded off: maximization of communication success (Equation 1) and minimization of energy exertion (Equation 2). Let us denote the set of intents (goals) to be communicated $G$ and motion trajectory $\tau \in T$, the set of all trajectories. We employ an $L2$ torque loss where $I$ is the moment of inertia and $\omega$ the angular velocity, for all agent joints. The total loss is a linear combination of prediction (cross-entropy) and energy (torque) losses.

$$L_{pred} = -\log p_\phi(\hat{g} = g^* \mid \tau) \tag{1}$$

$$L_{engy} = \| I * (\hat{\omega}_{1:T} - \hat{\omega}_{0:T-1}) \|_2^2 \tag{2}$$

The latent structure is implicit because agents are never made aware that any predefined, exploitable structure exists, nor are they directly incentivized to discover a more compact representation. This means if agents fail to *autonomously discover* the induced structure, they will be relegated to overfitting to trajectory input with their current partners and though they might perhaps find some structure, it will be insufficient for ZS coordination with novel partners. For this reason, in our experiments, we examine two ways of giving input to observer models: (1) provide actor trajectories generated, as this ideally is the goal, that the observer can autonomously discover any latent structure and exploit it for decoding messages, and (2) directly provide the latent energy values, as this allows decoupling of successfully induced structure in the policy learning from the successful discovery of that structure for the interpretation of messages. Both are necessary for successful coordination.

## 4.2 Providing Explicit Latent Structure

For explicitly providing structure, we define a set of latent features on trajectories $\Phi(T)$. This structure is intended to reduce dimensionality of messages passed, while preserving informativeness of messages. Thus we aim to induce a relationship $I(G; \tau) = I(G; \Phi(\tau)) \gg 0$, to achieve high mutual information between $G$ and $\Phi$. This relationship can be expanded as $I(G; \Phi) = H(G) - H(G|\Phi)$. However, the intent distribution is given as a Zipfian, so $H(G)$ is held constant. Thus, as is generally true in protocol learning, maximizing mutual information between goals and messages implies:

$$\min H(G|\Phi) \implies \begin{cases} p(g|\Phi(\tau)) \to 1 & g = g^* \\ p(g|\Phi(\tau)) \to 0 & else \end{cases} \tag{3}$$

This implication is critical for ZS coordination, in the absence of data to bias how agents learn to generate trajectories. It suggests that even if a new observer does not know how a particular actor communicates, i.e., it has not learned a mapping between actor trajectories and intents, *if* it can successfully infer the latent structure for encoding trajectories, it can still decode the message.

## 4.3 Third-Party Observer Evaluation

We evaluate policies learned using a third-party (external) observer, not trained on the protocol. Learned policies are frozen and the external observer initialized randomly. The population is split into disjoint training and test sets. At each iteration, it trains by *only observing* actors in the training set, and tests on its ability to correctly interpret intent of actors in the test set. Since all actors are trained *independently*, the ability to understand these unseen partners represents ZS communication.

## 5 Experiments and Results

Our experiments analyze: (1) the value of inducing implicit latent structure on embodied protocol learning, (2) the impact of discovery of the latent structure by observer agents – in the ZS coordination setting, and (3) challenges associated with optimizing in a continuous space for a monotonic ordering on energy values. We also provide some qualitative insights about learned policies. The experimental setup is detailed in the Appendix.

### 5.1 Efficacy of Inducing Implicit Structure in Protocol Learning

This first set of experiments in Figure 2 reports communication success during training, on a task with 10 intents. Results for our experimental condition (Energy + Zipf) are compared against all three ablations. Figure 2a shows all conditions successfully converge on a protocol with their training partners (SP), though conditions trading off two objectives converge more slowly. Figure 2b shows communication success for each actor paired with *unseen* observer agents in the population (all <actor, observer> pairs that did *not* train together) (CP). This represents an out-of-distribution test case. It shows that none of the models are able to immediately generalize to new partners (at test time), elucidating the need for additional third-party observer training, to increase the likelihood of communication generalization. It also shows there is value in using a nonuniform distribution, as it enables independent agents to agree a priori on how to prioritize intents for future interactions. Lastly, from this experiment, we found that while adding an energy objective does slow convergence, it reduced energy exertion by several orders of magnitude, which can be critical in reducing resource consumption and enabling prolonged operation for practical embodied systems (*e.g.* robots).

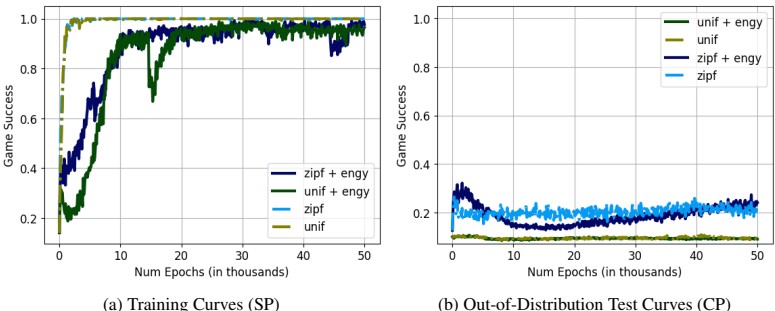

(a) Training Curves (SP)                    (b) Out-of-Distribution Test Curves (CP)

Figure 2: Learning Curves, associated with the four experimental conditions described. Subfigure 2a shows communication success with training partners (SP). Subfigure 2b shows communication success of each actor with all *unseen* observers in the population (CP), as actor policies are being trained. The latter is out of distribution, since all (actor, observer) pairs train *independently*. Plot illustrates that while protocol *training* converges to near perfect performance, *generalization* of protocol to novel partners is extremely challenging.

### 5.1.1 Efficacy of Implicit Structure in Discrete-Channel Protocols

To further investigate the value of coupling a Zipfian intent distribution with an energy cost objective for learning ZS policies, we evaluate a simple discrete-channel communication domain. In this simplified domain, we consider protocols with 5 intents and examine two learning tasks. In the first task, there are 10 discrete actions each sender agent can take, each assigned its own unique energy value: $\Phi(a) = id(a)$. Examples of learned protocols for this task are visualized in Figure 3. For achieving *both* maximal prediction accuracy *and* minimal energy exertion, each intent must be mapped to the lowest energy value it can be assigned, but no two intents can be mapped to the same action. The bottom protocol in Subfigure 3b illustrates a globally optimal solution for this domain in the ZS setting, where discriminability is maximized and energy minimized.

For quantitative analysis on the value of inducing energy costs on top of a Zipfian distribution, we trained five independent sender-receiver pairs, both without an energy penalty and using the penalty to induce the energy-based latent structure. For both conditions, we computed SP and CP performance between and across agent pairs, respectively. While SP performance at the end of protocol training is comparable for the conditions, CP performance is $0.15 \pm 0.12$ and $0.58 \pm 0.22$ for the zipf (only) condition vs zipf + energy conditions, respectively. Although this suggests significant variance in performance across pairs for CP evaluation, the key finding is still clear. Additional results for both learning tasks in this domain are discussed in Appendix Sections A.4 and A.5. We also contribute a colab notebook as an instructive tool for exploring protocol learning in this simple domain[1]. Overall, findings are consistent across tasks: coupling a Zipfian distribution with the energy objective is important for learning more *general* (ZS) communication skills. We continue our continuous-action protocol experiments using the implicit energy-based latent structure.

## 5.2 Impact of Latent Structure for Zero-Shot Coordination

While the previous set of experiments highlight the potential value of inducing the proposed implicit latent structure, we now analyze a *proof-of-concept* task, to extract insights about solving the problem setting of ZS coordination using a high-dimensional continuous communication channel. Table 1 shows ZS results given by the external observer evaluation, for a $N = 2$ concepts task.

| $N = 2$ Intents Task | | | |
|---|---|---|---|
| | | Test Input | |
| | | $\tau$ | $\Phi(\tau)$ |
| Train | $\tau$ | 0.75 | **0.97** |
| Input | $\Phi(\tau)$ | 0.67 | **0.997** |

| $N = 2$ Intents Task | | | |
|---|---|---|---|
| | | Test Input | |
| | | $\tau$ | $\Phi(\tau)$ |
| Train | $\tau$ | 0.66 | **0.995** |
| Input | $\Phi(\tau)$ | 0.67 | **0.999** |

(a) No Curriculum                    (b) Torque Curriculum

Table 1: CP with External Observer. *Proof-of-Concept* Task: 2 Concepts. In both settings (with or without a curriculum), near-perfect ZS coordination can be achieved.

[1]Colab notebook for experimenting with simple Discrete Domain – http://shorturl.at/luHPX

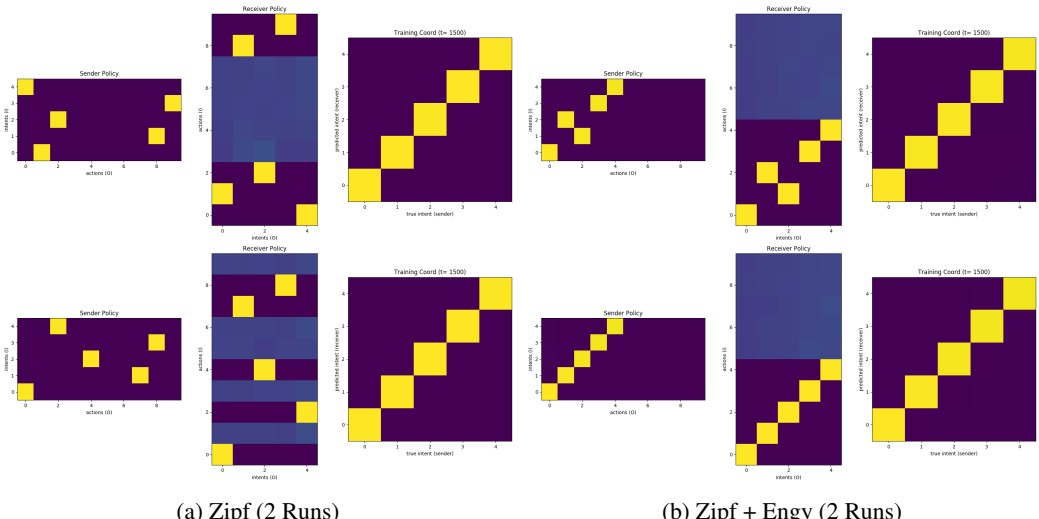

(a) Zipf (2 Runs)  (b) Zipf + Engy (2 Runs)

Figure 3: Protocols Learned in Discrete-Channel Domain. Two independently trained agent pairs (top, bottom) per condition. Left column shows sender policy mapping given intents to communication actions (messages). Middle column shows receiver policy mapping sender actions to predicted intents. Right column shows SP Performance ($p(predicted - intent \mid true - intent)$) at the end of Protocol Training. Illustrates both conditions can train sender policies to communicate effectively with training partners, but only the condition with the energy-based structure (3b) produces policies sufficiently similar for generalization *beyond* training partners.

In the tables, rows represent the type of input given to the observer model at training time and columns represent the type of input given at test time. Inducing a Zipf distribution on the intents, the likelihoods are $[0.67, 0.33]$ on concepts 1 and 2, respectively. Thus using a max class classifier as a reasonable baseline for comparison, we would expect a success rate of 0.67. Bold figures highlight conditions that achieved communication generalization, success beyond that of the baseline. The results show near perfect ZS coordination *only* when providing the external observer with the explicit latent structure (energy values), independent of what information was provided during training.

To gain more insights about why agents are able to generalize so well, given the latent energy variable, we visualize a distribution over energy values for $N = 2$ intents, for the best performing agent (Figure 4). Only given energy values at test time, the only way for an external observer to perform with such high accuracy is with very little overlap between the intent-conditioned Gaussian distributions. Looking at Figure 4, this is exactly what we see. The left plot shows the condition given trajectory input at training time and the right plot given the latent variable input. The trend is consistent and very clear. Given only these energy values at test time, an external observer can successfully decode the message, independent of the way the specific actor qualitatively communicates through motion. This means the implicitly induced latent structure successfully creates a maximally informative relationship between intent and energy exertion, as characterized by Equation 3.

Figure 5 illustrates snapshots of sample behaviors learned for one agent in the population on the N=2 task. They are visualized using the FAIR Motion Library (Gopinath & Won, 2020). From this visualization, we observe *qualitatively* distinct communicative behaviors for the two intents.

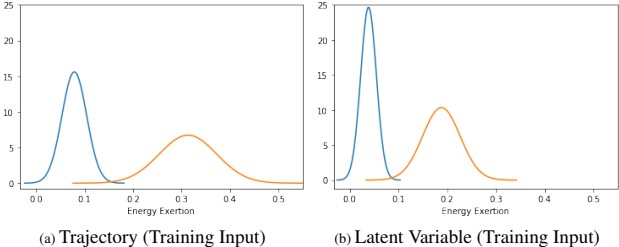

(a) Trajectory (Training Input)  (b) Latent Variable (Training Input)

Figure 4: Energy Exertions of Intents 1 (blue) and 2 (orange). Task = 2 Intents. Shows intent-conditioned Gaussian distributions to approximate energy exertion of one agent, given both types of Training Input. *High* ZS coordination is achieved in this task because there is *minimal overlap* between the intent energy distributions.

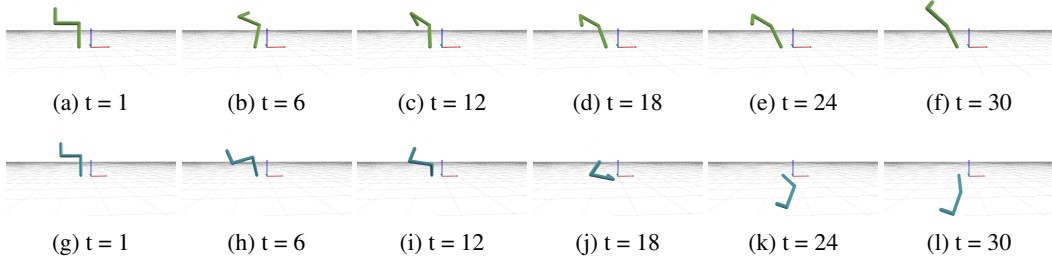

| (a) t = 1 | (b) t = 6 | (c) t = 12 | (d) t = 18 | (e) t = 24 | (f) t = 30 |
| (g) t = 1 | (h) t = 6 | (i) t = 12 | (j) t = 18 | (k) t = 24 | (l) t = 30 |

Figure 5: Comparison of Learned Behaviors for N=2 Intents Task. Illustrates one instantiation of how an agent learns to generate communication for distinct intents in the protocol. Top = Intent 1, Bottom = Intent 2. Note: Although the time horizon is $T = 5$ for policy learning, trajectories are upsampled for visualization purposes to $T = 30$. So visualized snapshots are evenly spaced throughout an entire generated trajectory.

The primary insight from this proof-of-concept experiment is that we *can* indeed achieve ZS coordination with this energy-based implicit structure underlying the protocol. However, coordination with novel partners remains improbable *unless* the latent structure is first discovered by the agents.

## 5.3 Improving Communication Generalization with a Learning Curriculum

| $N = 5$ Intents Task | | Test Input | | | $N = 5$ Intents Task | | Test Input | |
| --- | --- | --- | --- | --- | --- | --- | --- | --- |
| | | $\tau$ | $\Phi(\tau)$ | | | | $\tau$ | $\Phi(\tau)$ |
| Train | $\tau$ | 0.37 | **0.76** | | Train | $\tau$ | 0.44 | **0.70** |
| Input | $\Phi(\tau)$ | 0.44 | **0.60** | | Input | $\Phi(\tau)$ | 0.44 | **0.70** |

(a) No Curriculum          (b) Torque Curriculum

| $N = 10$ Intents Task | | Test Input | | | $N = 10$ Intents Task | | Test Input | |
| --- | --- | --- | --- | --- | --- | --- | --- | --- |
| | | $\tau$ | $\Phi(\tau)$ | | | | $\tau$ | $\Phi(\tau)$ |
| Train | $\tau$ | 0.30 | **0.61** | | Train | $\tau$ | **0.56** | **0.56** |
| Input | $\Phi(\tau)$ | 0.34 | **0.55** | | Input | $\Phi(\tau)$ | 0.35 | **0.56** |

(c) No Curriculum          (d) Torque Curriculum

Table 2: CP with External Observer. For tasks of increasing complexity: 5 and 10 intents. ZS Coordination is maximized when using an energy curriculum and providing the explicit latent structure.

Observing such high ZS coordination success on the proof-of-concept task is very promising. However, we also aim to characterize performance as the complexity of the communication task increases. With this in mind, we investigated generalization with an external observer on tasks with 5 and 10 communicative intents, respectively. The left side of Table 2 (2a and 2c) shows ZS coordination results after training protocols with the implicit latent structure. The most likely intents are sampled with a likelihood of approximately 0.44 and 0.34 for the $N = 5$ and $N = 10$ concepts tasks, respectively; so these probabilities form baselines for comparison against a naive max class classifier. Bold figures highlight conditions with *some* communication generalization. In the case of the $N = 5$ task, we see a consistent trend as observed in our proof-of-concept task: the external observer can successfully generalize to unseen actors only when given the explicit latent structure. Though notably, ZS coordination success is substantially diminished with this more difficult communication task. Consistently for all three tasks, when *given trajectory input*, the external observer cannot outperform simply selecting the most likely intent. Furthermore, with an increase in task complexity to 10 concepts, the external observer *never* performs better than this baseline strategy.

In analyzing these trends more closely, it is important to consider that as the number of concepts increases, the number of ways to order the corresponding energy values increases combinatorially. It follows that as the complexity of the task increases, the optimization algorithm becomes more susceptible to getting stuck in a local optimum. We conduct analysis on optimizing with the energy objective, to better understand the behavior of the energy optimization and accordingly decide how to address the most salient challenges arising. Figures 6a and 6b show the energy loss over time on the $N = 5$ concepts task, given both types of input. The plots subsample energy loss values over the

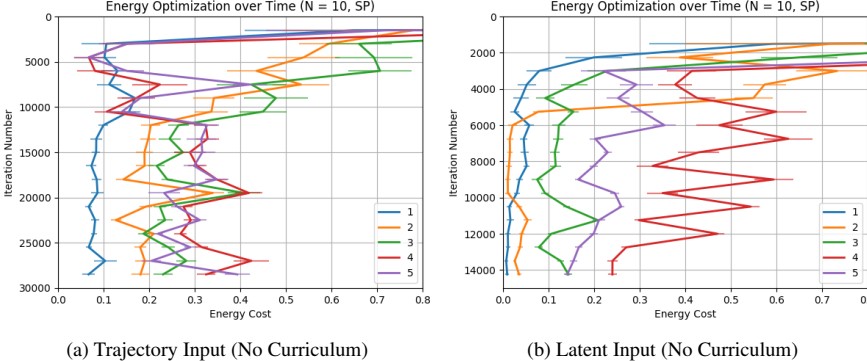

(a) Trajectory Input (No Curriculum)     (b) Latent Input (No Curriculum)

Figure 6: Energy Costs by Intent, sub-sampled throughout duration of protocol training. Energy cost for each intent proceeds over time – from top to bottom. Task = 5 concepts. Training with *no* curriculum. Shows *correctly* mapping intents by rank to continuous-valued energy costs is challenging for the optimization algorithm. Even *sufficiently good* solutions converged upon can still be *suboptimal* (see intents 4 and 5, subfigure 6b).

entire duration of training, for each run. Each intent is represented by a different colored line and shaded horizontal lines show the variance of the energy loss, at a particular point in training.

A key observation from this analysis is repeated repositioning of intent ordering as training traverses, and there is noticeably more repositioning of intents when trajectory input is given. This can be explained by the dimensionality of the trajectory input being orders of magnitude larger than the scalar latent input; thus there are many more degrees of freedom in the search space for the optimization algorithm to manipulate the positioning of the intents. Yet for both input types, there is very little repositioning of intent 1. The Zipfian imposes the following approximate distribution over five elements: $[0.44, 0.22, 0.14, 0.11, 0.09]$. Notably, the difference between intent 1 and any other intent is much larger than the difference between any other pair of intents. This suggests that the optimization algorithm would incur too much cost if it orders intent 1 incorrectly, so energy cost of intent 1 generally remains lowest throughout training. By contrast, the positions of much less likely intents (*e.g.* 4, 5) continue to shift (particularly given trajectory input) and at many points during training, are incorrectly ordered as the optimization algorithm searches for a *sufficiently good* solution. This implies that the constant shifting of intent-conditioned energy values is bound to impact coordination between actor and observer, since it influences trajectories generated for communication.

From this insight, we decided to pretrain actor policies on energy, by first minimizing torque in trajectory generation for *all* intents uniformly. Our hypothesis was that training the protocol with an energy-based curriculum may help to reduce the amount of intent repositioning necessary to find the optimal ordering, since the intents will all start with comparable energy exertions. The right side of Table 2 (2b and 2d) shows ZS coordination results after training protocols with the latent structure *and* torque-based curriculum. The torque curriculum seems to only improve generalization notably when the external observer is given trajectory input for the hardest communication task ($N = 10$).

Overall, two key insights emerge from our experiments and analyses: (1) it is advantageous to *induce* an energy-based latent structure for learning generalizable communication protocols and (2) explicit *discovery* of the latent structure is a separate yet critical piece for ZS coordination to occur.

## 6 CONCLUSION AND FUTURE WORK

We have presented a first exploration into emergent non-verbal communication in the context of embodied agents in high-dimensional simulated environments. We show that under mild assumptions, the grounding provided by the physical environment should allow our agents to learn protocols that can generalize to novel partners. Yet we also find that the current approaches are more brittle than one might hope. We hypothesize that better robustness could be achieved by methods that can discover the maximum possible coordination from a given level of common knowledge ('grounding') in the environment. Specifically, this may require distinguishing between those aspects of the optimization problem that are idiosyncratic to a given agent (or might even be shared knowledge) compared to those that are common knowledge (and can therefore be used to coordinate). Overall, while there are many interesting open challenges that remain, this work opens up exciting avenues for exploring continuous action communication protocols in virtually or physically embodied agents.

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

## A    APPENDIX

### A.1    EMBODIED AGENT MORPHOLOGIES

Embodied agents explored in this work have articulated bodies. An articulated body is collection of rigid bodies (links), connected by a set of joints $j \in J$. The body of an agent is constrained kinematically, such that links cannot detach from one another. Topology defines a tree structure for the body, where each joint is a node and edges denote a link between joints. The root joint, $j_0$, is the root of the tree. Only the root joint has six degrees of freedom (DOF): $\langle p_x, p_y, p_z, r_x, r_y, r_z \rangle$. Since the body must move as one entity and thus links cannot detach from one another, given no further joint limits or constraints, all other joints have the ability only to *rotate* freely in Euclidean space. So all derivative (child) joints $j_i$ where $i \neq 0$ have three DOFs: $\langle r_x, r_y, r_z \rangle$. For simplicity, we constrain all agent joints with only the DOFs to rotate in three-dimensional space. This means agents *cannot navigate* through the environment; their bodies can only move *in place*. Moreover, a predefined topological structure implies trajectories generated must adhere to the kinematic constraints of the agent given a priori. We employ a forward kinematics (FK) transition model to ensure valid transitions between joint states. The FK module is adapted from (Holden et al., 2017) to be differentiable.

## A.2 PROTOCOL TRAINING

---
**Algorithm 1** Training Overview
---
1: Let $G \leftarrow$ set of communicative intents (goals)
2: Let $A \leftarrow$ population of agents
3: Randomly Initialize $\Theta, \Phi$   #(policy and discriminator models, for all agents)
4: **for** $i \in Iterations$ **do**
5:    # Train Sender $\theta$ + Receiver $\phi$
6:    $B \sim G$   #(sample batch of intents, according to predefined distribution)
7:    **for** $g \in B$ **do**
8:       $s_0 \leftarrow P_0$   #(agent reference pose)
9:       **for** $t \leq T$ **do**
10:          $\mu(s, \theta) \leftarrow \pi(s_t \| g)$   #(compute policy action - given state and target intent)
11:          $a_t \leftarrow \mu(s, \theta) + N(0, I) * \sigma$   #(add noise - for exploration and imperfect actuation)
12:          $s_{t+1} \leftarrow FK(s_t, a_t)$   #(analytically compute next state - using *given* agent kinematics)
13:       **end for**
14:       $\tau \leftarrow \{s_1, a_1, s_2, a_2, \ldots, s_T\}$   #(compose actor trajectory - sequence of joint states and actions)
15:       $\hat{g} \leftarrow D_\phi(\tau)$   #(compute observer prediction of intent)
16:       $R = -L(g, \hat{g})$   #(compute shared return)
17:       Update model parameters $\Theta, \Phi$
18:    **end for**
19: **end for**
---

## A.3 EXPERIMENTAL SETUP

We design an *Intent Recognition* referential task for our experiments and consider a library of 10 communicative intents (concepts) given a priori (*e.g.* "yes", "no", "turn left", "come here"). Each population is instantiated with a size of 10 agents. The agents are not situated in any physical or virtual environment. At each iteration of the game, a batch of 1024 intents is sampled, based upon the specified distribution (uniform or Zipf). Each sender agent generates a batch of 1024 intent-conditioned trajectories. Receiver agents are able to directly observe positions and orientations of sender joints. However, noise is added to each action, to represent imperfect observability. Noise emission is modeled through a unimodal Gaussian distribution $N(0, \sigma)$, with fixed $\{\sigma_p = 2.0, \sigma_r = 0.4\}$ values for joint positions and rotations, respectively. Here, position values are assumed to be measured in feet and rotation values in radians. Both policy and discriminator models are 3-layer feed-forward neural networks, with ReLU activations on all layers except the final. The discriminator network uses a SoftMax for the final layer, and the policy network uses no activation in the final layer, since it is allowed to output a broad range of continuous action values.

We experiment with a robot arm morphology in this work, motivated by potential application domains where non-verbal communication is most relevant, *i.e.* robotics and graphics. The agents in the population each have 4 articulated joints. No constraints were imposed on agent joints, so as to allow maximal range of motion. We did however employ action equivalences, whereby the modulus operator was applied to joint angles predicted by the policy network; this confined all joint angle values to the range of $[0, 2\pi]$. We employed the same learning rate for both the policy and discriminator networks, $1e - 3$. As a simplifying assumption, we consider a small time horizon (T = 5), but upsample the number of frames for visualization.

For the external observer evaluation, we randomly initialize a new discriminator model and split the original community, with 80% of the agents assigned to training set and 20% assigned to the test set. In each iteration, the training actors *each* generate a batch of 1024 intent-conditioned trajectories, based upon the sampled Zipf intent distribution. The external observer is also tested at each iteration with the trajectories generated by the test actors, who also each generate a batch of 1024 trajectories. The external observer experiments were run for 500 iterations on $N = 2$ and $N = 5$ concepts tasks and 1000 iterations on the $N = 10$ task, though learning stabilizes after approximately 100 iterations.

## A.4 DISCRETE DOMAIN: TASK 1 – ADDITIONAL RESULTS

The first task in this simple discrete-channel domain considers 10 discrete actions each sender agent can take, each assigned its own unique energy value: $\Phi(a) = id(a)$. Examples of three independently

trained protocols for baseline (Zipf *only*) and experimental (Zipf + Energy Penalty) conditions of this task are visualized in Figures 7a and 8a, respectively. Visual illustrations of pairing off the senders with unseen receivers from the set in CP (for evaluation of ZS coordination) are shown in Figures 7b and 8b. These are juxtaposed with SP pairings and protocols trained (Figures 7a and 8a).

## A.5    DISCRETE DOMAIN: TASK 2 (PROTOCOL LEARNING GIVEN ACTION DEGENERACIES)

The second task in the simple discrete communication domain also considers protocols with 5 intents and only five distinct energy values. However, there are now 17 discrete actions, which implies action redundancy. In particular, one action has a unique, lowest energy value; the 4 other energy values are each mapped to four different actions.

In Physics, a degeneracy denotes the number of distinct quantum states of a system that have a given energy. Analogously, this experiment investigates what happens in the degenerate case where several actions can be mapped to the same energy cost. Thus even in optimizing for minimal effort protocols, there are many ways to converge upon a globally optimal protocol, and consequently, ZS coordination becomes more challenging. This is because a sender-receiver pair can *both* maximize disriminability *and* minimize energy by assigning different actions (but with the same low energy exertion) to different intents.

This gets closer to challenges faced with continuous action protocols, where as the likelihoods of different intents grow arbitrarily close together, a sender may decide to employ visually distinct trajectories, but with comparable (low) energy exertions, to different intents. When this happens, it becomes exceedingly difficult to differentiate intents (for a new receiver agent) based upon the latent energy values, and ZS communication performance degrades.

Examples of learned protocols for baseline (Zipf *only*) and experimental (Zipf + Energy Penalty) conditions of this task are visualized and zoomed in on in Figures 9 and 10, respectively. Those same protocols are juxtaposed with visual illustrations of pairing off senders with unseen receivers for CP, in Figures 11 and 12.

For quantitative analysis on the value of inducing energy costs on top of a Zipfian distribution, for both conditions, we computed SP and CP performance between and across agent pairs, respectively. Like the simpler (first) task in this discrete domain, while SP performance at the end of protocol training is comparable for the two conditions, CP performance is notably different between them. In particular, CP success was $0.20 \pm 0.07$ and $0.35 \pm 0.2$ for the zipf (only) condition vs zipf + energy conditions, respectively. As an additional point of comparison, the most frequently occuring class for $N = 5$ intents occurs with approximately 0.44 probability. So even in the experimental condition which performed better, ZS coordination capability does not exceed simply guess the maximally occurring class; this speaks to the inherent difficulty of learning ZS policies, even in seemingly simple domains. The empirical findings show evidence that on this harder communication task (with action degeneracies), ZS coordination is more difficult to achieve than as compared to the simpler task where all actions were assigned unique energy values. This was expected, but still useful to confirm. Lastly and importantly, these findings suggest that even when action degeneracies exist, coupling a nonuniform distribution with the energy objective is helpful for maximizing *generality* of communication skills learned. This finding is also consistent with what we observed on the simpler task in this discrete communication domain.

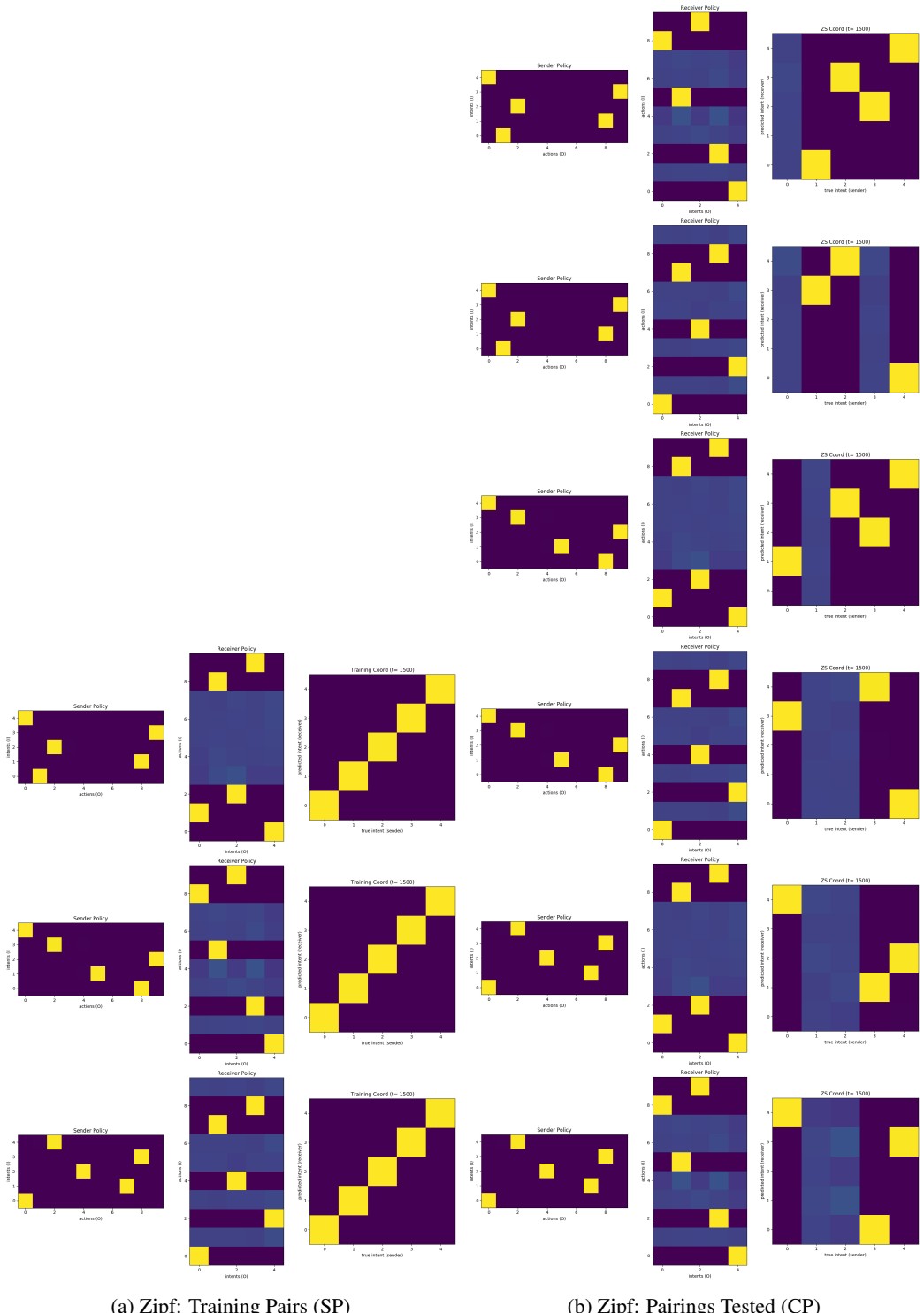

(a) Zipf: Training Pairs (SP)   (b) Zipf: Pairings Tested (CP)

Figure 7: Protocols Learned and Tested for ZS Coordination in Discrete-Channel Domain (Task 1). **Zipf *only* Baseline Condition**. Protocol Training (SP) and Evaluation (CP). Illustrates how challenging ZS communication (7b, right column) is as compared to communication success with training partners (7a, right column). It is made harder by the fact that policies learned during the initial protocol training phase are very different, across senders (7a, left column)

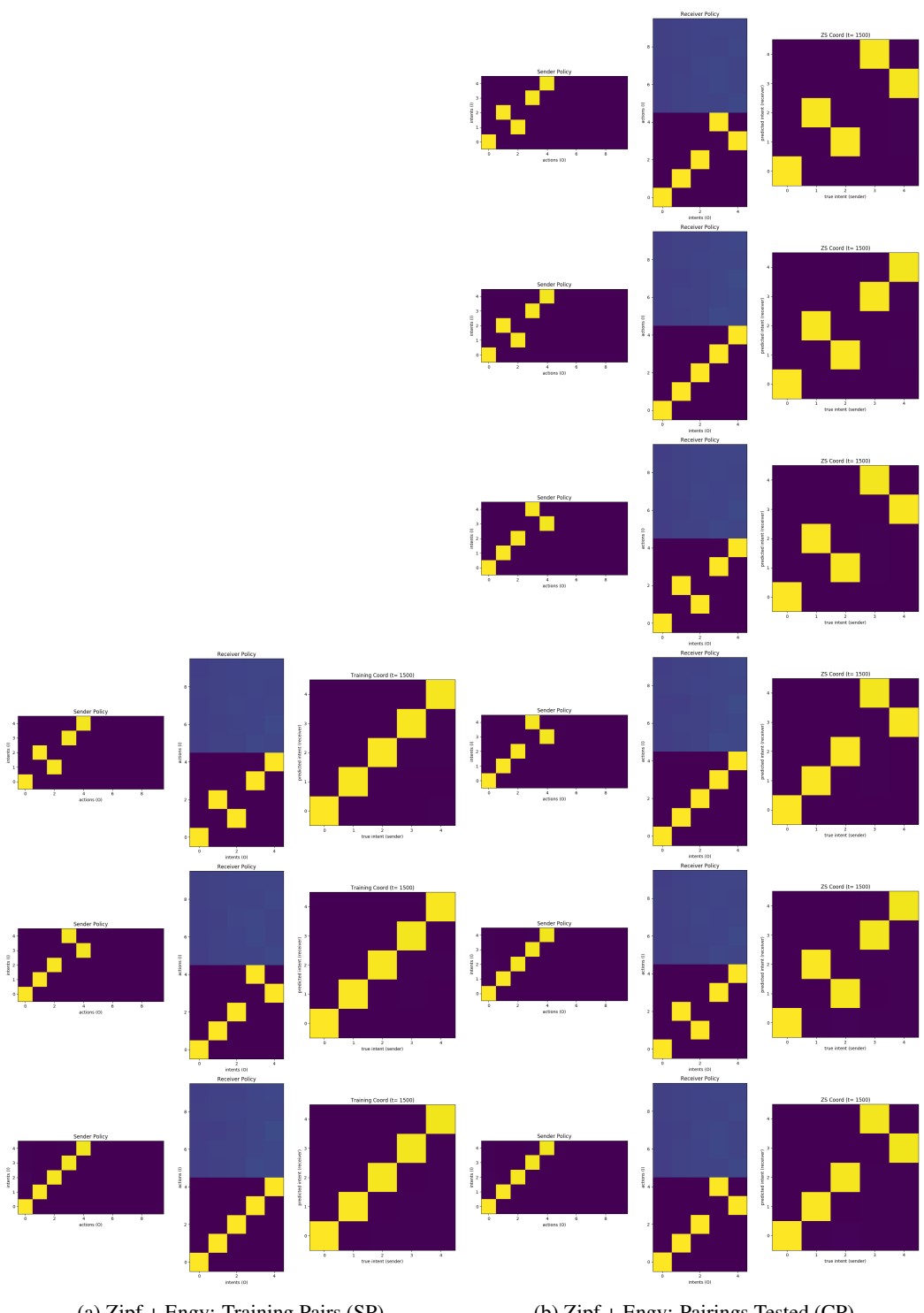

(a) Zipf + Engy: Training Pairs (SP)  (b) Zipf + Engy: Pairings Tested (CP)

Figure 8: Protocols Learned and Tested for ZS Coordination in Discrete-Channel Domain (Task 1). **Zipf + Energy Experimental Condition**. Protocol Training (SP) and Evaluation (CP). Illustrates that while ZS communication (8b, right column) is somewhat more challenging than success with training partners (8a, right column), ZS coordination from protocols built upon energy-based latent structure is noticeably more successful than ZS coordination in baseline condition with no energy penalty (8b, right column). This can be observed by higher prediction probabilities (shown as yellow boxes) occurring with significantly greater frequency along the diagonal (meaning sender and receiver agree on intent) than in baseline condition.

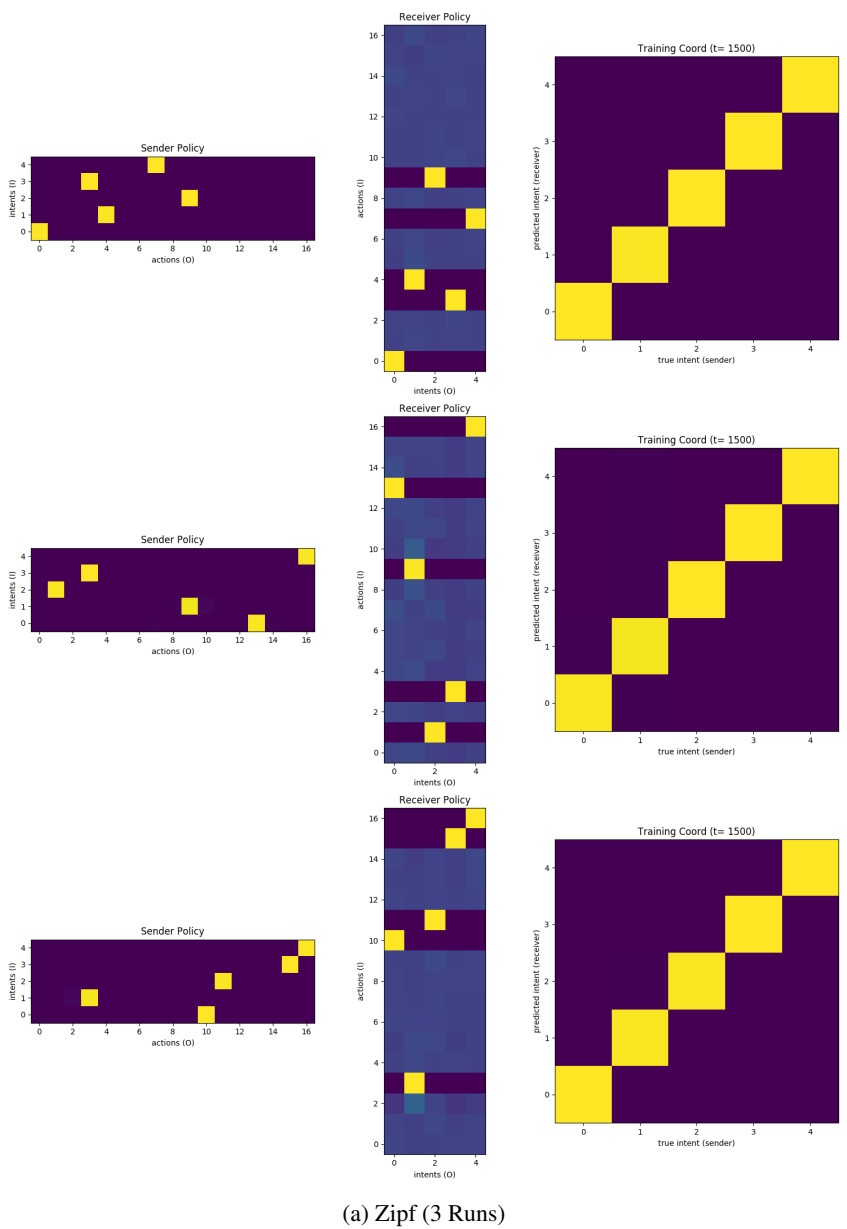

(a) Zipf (3 Runs)

Figure 9: Protocols Learned in Discrete-Channel Domain, with Action Degeneracies. Three independently trained agent pairs (individual rows) in baseline condition. Left column shows sender policy mapping given intents to communication actions (messages). Middle column shows receiver policy mapping sender actions to predicted intents. Right column shows SP Performance ($p(predicted-intent \mid true-intent)$) at the end of Protocol Training. Illustrates condition can train sender policies to communicate effectively with training partners, but there is significant variance across sender policies learned. This would make finding common structure and exploiting it to generalize to novel partners very difficult (perhaps impossible since no common structure is evident).

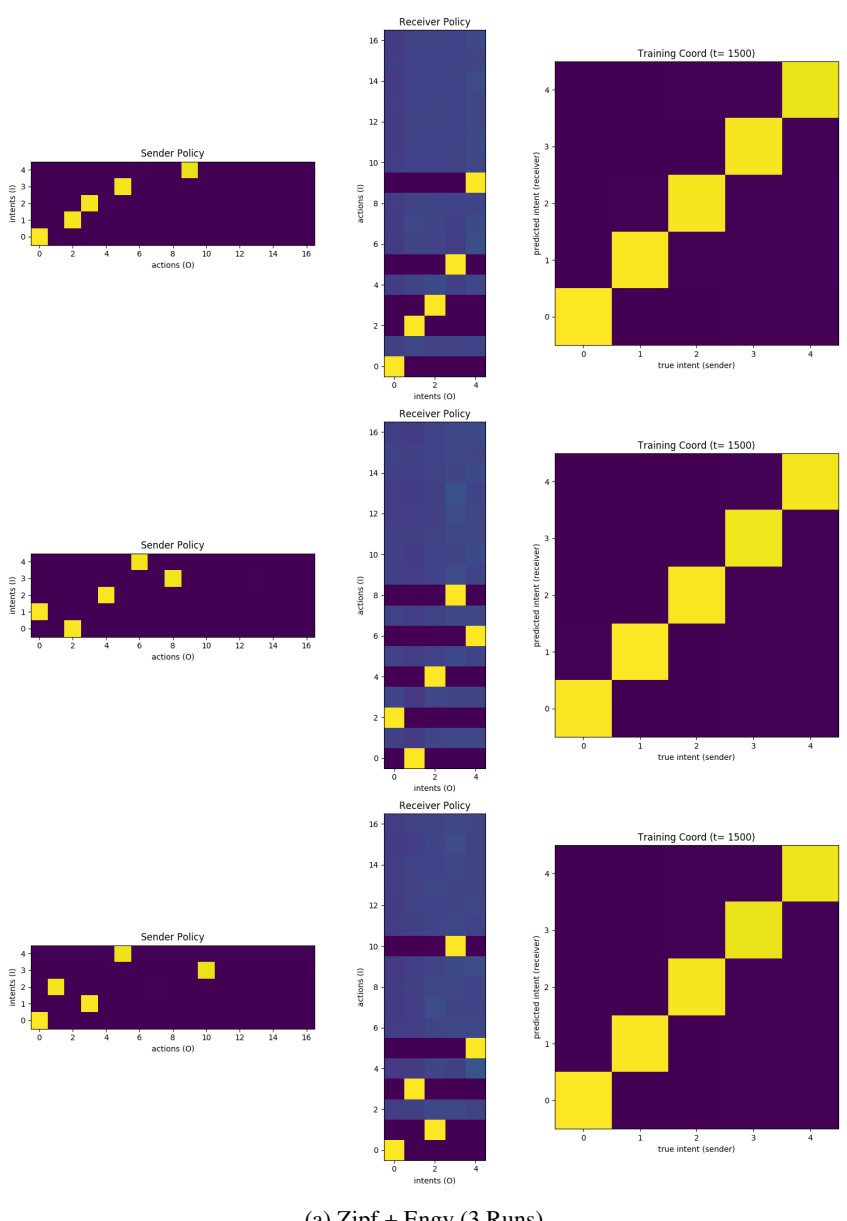

(a) Zipf + Engy (3 Runs)

Figure 10: Protocols Learned in Discrete-Channel Domain, with Action Degeneracies. Three independently trained agent pairs (individual rows) in experimental condition. Left column shows sender policy mapping given intents to communication actions (messages). Middle column shows receiver policy mapping sender actions to predicted intents. Right column shows SP Performance ($p(predicted-intent \mid true-intent)$) at the end of Protocol Training. Illustrates condition using energy-based latent structure can both train effective protocols between training partners and induce similar structure in policies learned, *across* sender agents, for enabling better communication generalization (to unseen partners).

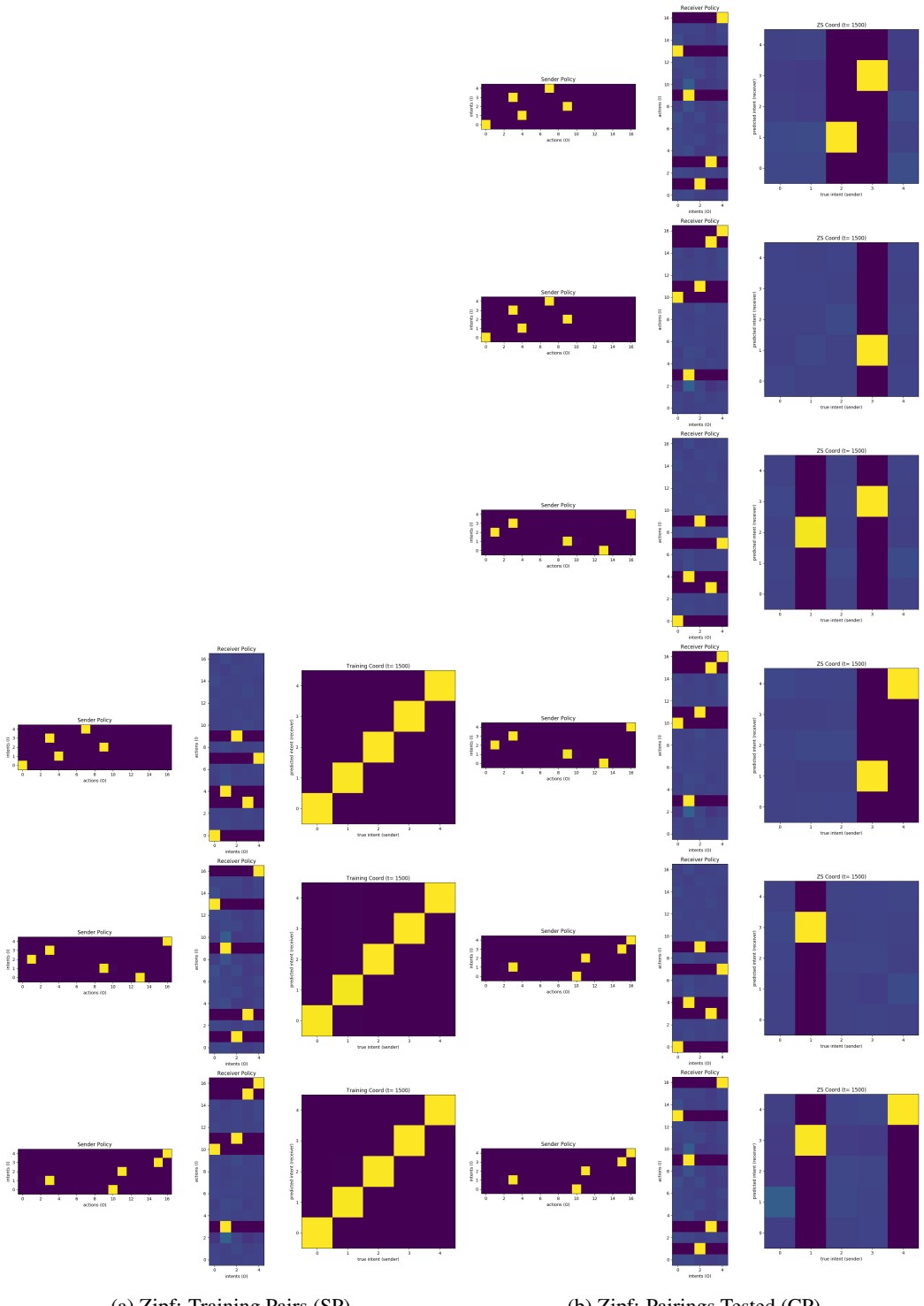

(a) Zipf: Training Pairs (SP)  (b) Zipf: Pairings Tested (CP)

Figure 11: Protocols Learned and Tested for ZS Coordination in Discrete-Channel Domain – given Action Degeneracies (Task 2). **Zipf *only* Baseline Condition**. Protocol Training (SP) and Evaluation (CP). Illustrates how challenging ZS communication (11b, right column) is as compared to communication success with training partners (11a, right column). It is made harder by the fact that policies learned during the initial protocol training phase are very different, across senders (11a, left column).

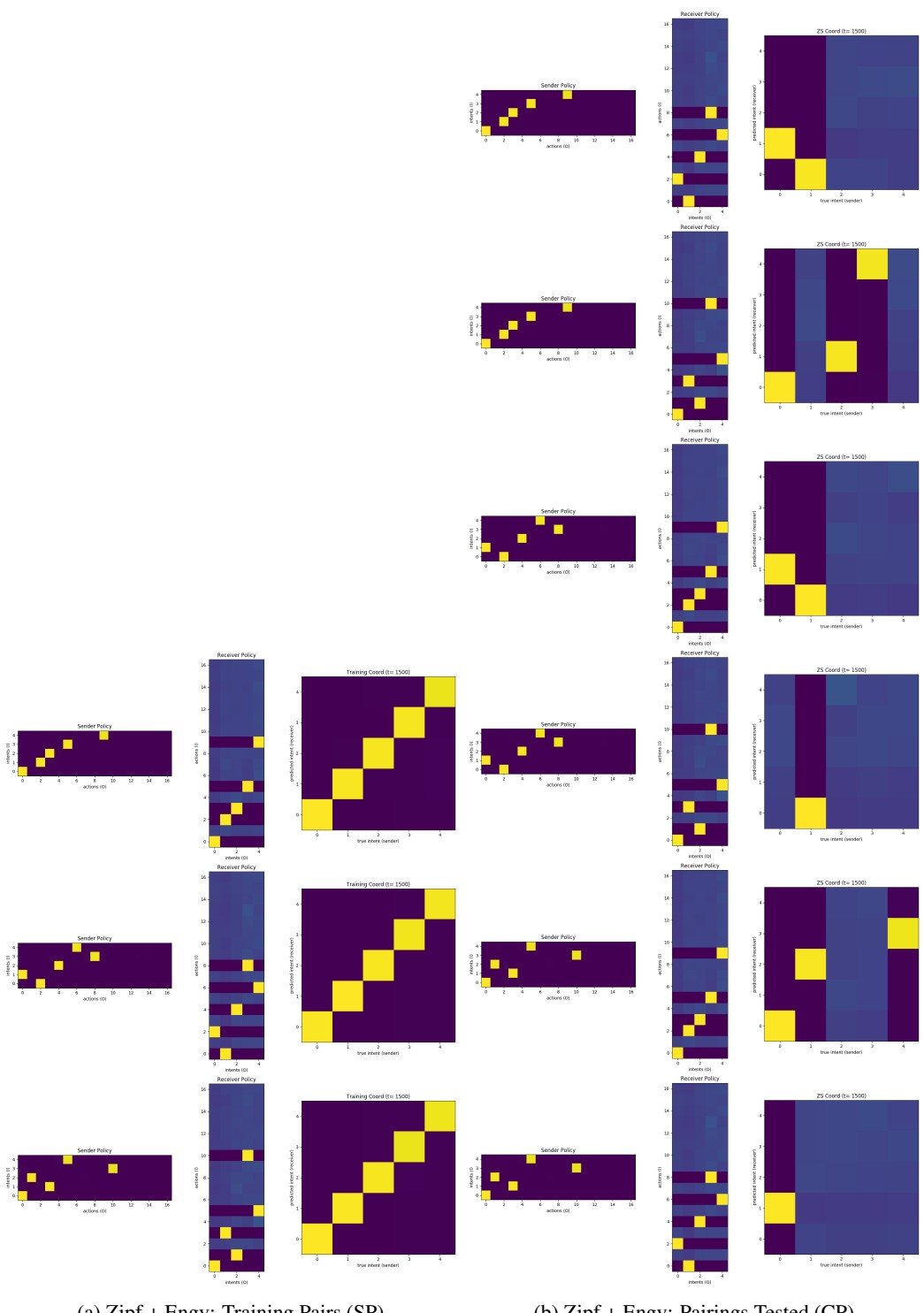

(a) Zipf + Engy: Training Pairs (SP)          (b) Zipf + Engy: Pairings Tested (CP)

Figure 12: Protocols Learned and Tested for ZS Coordination in Discrete-Channel Domain – given Action Degeneracies (Task 2). **Zipf + Energy Experimental Condition**. Protocol Training (SP) and Evaluation (CP). Illustrates ZS communication (12b, right column) is substantially more challenging than success with training partners (12a, right column). Nonetheless, ZS coordination from protocols built upon energy-based latent structure is slightly more successful as compared against ZS coordination in baseline condition with no energy penalty (12b, right column).

