# OpenReview forum: "Exploring Zero-Shot Emergent Communication in Embodied Multi-Agent Populations"
_ICLR.cc/2021/Conference — Reject_

### Official Review · AnonReviewer3 · 2020-10-14
**The idea of using energy cost for zero-shot coordination is interesting, but why it is preferred is missing.**

**Rating:** 6
**Confidence:** 3

**Review:**

This paper studies how gesture-based (non-verbal) communication can emerge in embodied multi-agent population.
In their problem setting, there is a set of agents, and an observer. Given an intent sampled from a Zipf distribution, the agents generate energy-efficient trajectories, and the observer predicts the intent given these trajectories. Zero-shot coordination is enabled by combining energy cost with Zipf distribution (which is monotonic), since energy cost is grounded in the environment. The authors evaluate zero-shot coordination performance by evaluating a third-party observer trained and tested on two separate subsets of agents from the population respectively.

Pros:
1. The idea of using energy cost and combining energy cost with a non-uniform distribution for emergent communication is new.
2. The idea of exploiting common-knowledge grounded in the environment for zero-shot coordination is new and interesting.

Cons:
1. The paper is sometimes hard to follow with missing details.
2. The authors failed to discuss options other than energy cost for ZS coordination, and why energy cost is preferred compared with other options.
3. The assumption of a monotonic ordering for intent distribution is strong. The assumption that energy cost is available during training/testing is also strong.

I like the idea of exploiting a universal (common-knowledge) cost grounded in the environment for zero-shot coordination.
However, is energy cost the only option here? what about trajectory length (spatial), cumulative angular velocity change, etc. I understand energy cost may be a better option, but the authors should discuss and compare with other options. Meanwhile, in many multi-agent environment, the only available inputs are visual inputs and energy cost may not available. Finally, even if the learned policy achieved zero-shot coordination with a third-party observer, it is not human interpretable since humans cannot directly observe energy cost. Any thoughts on learneing human-interpretable gesture-based language?

While Figure 1 is "Overview of Learning System", it does not really helps the readers to get the key idea of the paper. It is better to have a more detailed figures explaining your problem setting and motivation.

The assumption that the intent is from a monotonic distribution seems a little bit strong to me. It is better to have experiments with other distributions (not uniform). e.g. a subset of intents has equal probabilities to be selected.

In Section 3 Problem Setting, you mentioned that "After T steps, the episode terminates, and the sequence of ...". So the number of steps per trajectory is fixed? Why?

In 5.1 where are the "unseen observer agents in the population" come from? My understanding is that there is one observer and a set of agents during the training. But here it implies that there are more than one observer during the training?

Algorithm 1 in A.2 is not clear. For example, Line 2 says "Let A<-population of agents". However, "A" never showed up again in the remaining of the algorithm; The term discriminator (line 3), receiver (line 5), and observer (line 15) are used interchangeably in Algorithm 1.

---

> ### Author Response · Authors · 2020-11-24
> **REBUTTAL (Reviewer 3)**
>
> We would like to thank you for your positive and constructive feedback.
>
> @the justification for energy cost as the grounding for ZS coordination: This is a good question.  Energy cost is certainly not the only option, but it represents a well-grounded and intuitive starting point for the investigation of learning generalizable protocols, where the modality for communication is physical movement of joints.  In particular, because the communication channel in this problem setting is both high-dimensional and continuous, we needed a principled way to reduce the expressivity for generating communication while preserving the informativeness of any messages passed.  Though we only use Zipf’s law [Zipf, 2016] for inducing a nonuniform distribution over communicative intents, it also highlights recurring patterns of minimizing effort for other human behaviors (e.g. with information-seeking).  Literature in Evolutionary Biology suggests this principle of least effort has been observed with animals as well.  It is analogous to the path of least resistance.
>
> So while there may be other options for grounding, it is important to remember this is (to our knowledge) a first investigation into communication protocols using movement of joints.  The goal then was to open up the space for exploration of this problem setting and contribute meaningful insights to the community.  Investigation of other common knowledge constraints or ways to ground the protocol in the environment could be quite interesting topics for future work.
>
> @learning human-interpretable gestures:  We find this to be a very interesting and relevant topic for investigation in future work.  However, it is important to note that in this work, our goal was to learn protocols that are self-compatible (agents effectively communicating with and generalizing to other agents).  The problem of learning human-compatible protocols or even evolving from learned self-compatible protocols to human-compatible protocols are certainly interesting research questions, but orthogonal to the current investigation.  And as embodied protocol learning represents a highly underexplored problem setting, learning self-compatible protocols is an important first step.
>
> @the assumption that the intent distribution is monotonic is strong: We agree that studying other intent distributions could be interesting for future work.  However, we’d like to point out that with the Zipfian distribution, as the number of intents grows, the lower likelihood intents begin to have very similar probabilities, so it closely approximates a subset of intent likelihoods being uniform.  It is also important to note that even with this perhaps strong (simplifying) assumption of monotonicity, finding a globally optimal protocol is still a really challenging problem.  We even observe this trend in our newest toy experiments, when learning protocols in simple discrete channel domains (new Section 5.1.1 and Appendix Sections A4 - A5).  So we believe it is important to first tackle this challenge, initially using reasonable simplifying assumptions.  As progress is made, assumptions can always be relaxed accordingly.
>
> @”...the number of steps per trajectory is fixed? Why?”:  T being fixed is a relatively standard assumption in continuous RL settings and in problem settings where learning motion policies.
>
> @“unseen observers in the population” referenced in Section 5.1: Thank you for the feedback about your confusion. Initial protocol training takes place between N actor-observer pairs being co-trained, so there are N actors and N observers at the end of protocol training.  Then for the third-party observer training, we freeze all actor (sender) policies and separate them into train and test sets.  A new external observer (receiver) is initialized randomly, trained with the training set of actors, and tested for ZS coordination with the test set of actors.  So in section 5.1, the out-of-distribution tests pair each actor with all observers it did not see during protocol training (Cross Play).  We have clarified this in the first paragraph of Section 3, when we discuss the Problem Setting. If accepted, we can also add a diagram to visually illustrate the overall method (protocol training + external observer training and evaluation).

---

### Official Review · AnonReviewer2 · 2020-10-28
**Introduces two interesting ideas at once, but this makes interpretation difficult**

**Rating:** 5
**Confidence:** 4

**Review:**

*** Summary ***

The paper investigates emergent gesture-based communication in Embodied Multi-Agent Populations. A noticeable feature of the paper is that it investigates emergent communication in the case of non-uniform distribution of intents and costly communication (i.e. agents are penalized for effort). The authors find that in certain scenarios, these conditions may lead to communication generalization of learned communication strategies to new, previously unseen agents.

*** Relevance, Originality, Novelty ***

I believe that the paper meets ICLR conference standards when it comes to the importance of the problem considered, the originality of the approach and the novelty of the proposed ideas.

*** Clarity ***

The paper is, in general, clearly written and is a pleasure to read. Figure 4 is deeply confusing, however. As mentioned in the appendix, the actual number of timesteps is equal to 5 and is upsampled during visualization. I believe that it is important to mention it right away (i.e. in the image description).
Otherwise, the reader can easily misunderstand the setup after seeing this picture.

*** Quality ***

The quality of the experimental support is, unfortunately, the weakest part of the contribution.

The authors introduce a number of exciting ideas, all of which make the communication learning setting more realistic. Specifically, the authors are studying communication in an embodied setting, Zipfian intent distribution, and the idea of effort-based action cost.

At the same time, while the ideas sound very promising in theory, in practice, the complications introduced by the embodied setting complicate the interpretation of the obtained results when it comes to the effects of Zipfian intent distribution and effort-based action costs.

In order to mitigate the difficulties, authors propose additional measures: providing explicit feature information to the outside observer (i.e. the observer directly gets the action effort); using "torque pretraining" to make the model more familiar with the reward landscape, dramatically reducing the number of actions.
These measures, in my opinion, are significantly detrimental to the potential impact of the paper. Providing effort information directly they makes the setting less realistic, defeating the purpose of the embodied approach.  Torque curriculum makes the results very specific to the considered setting.

There are other significant limitations of the experimental results that are probably indirectly induced by the embodied setting. Specifically, figure 2 and table 1 report no variability measures for the obtained result, and it's not clear, how many runs were made to obtain these results. Moreover, the agent population considered in these experiments is 10 (mentioned in the appendix). With such a small population, it is not surprising that the observer can simply memorize the individual patterns, and is not pressured to infer the underlying structure.  It seems very natural, therefore, to increase the number of agents in the simulation.
I suspect that these limitations are a consequence of the fact that training in the multidimentional setting is very costly and time-consuming.

Moreover, some of the limitations I describe in the "technical soundness" section could have been controlled for in a simpler setting.

Overall, all things considered, it's not clear what are the insights gained from the embodied approach.

*** Technical soundness ***

There are certain technical problems with the contribution:

Firstly, the justification for why Zipfian intents, together with effort-based penalty should lead to generalization is not rigorous (paragraph 1, section 4.1), and not completely convincing.

One of the sentences in the intuitive justification raises particularly serious concerns. The authors say "Coupled, [Zipfian intents and Energy regularization] incentivize an inverse relationship between energy exertion and intent frequency, assigning minimum energy trajectories to maximally occurring intents.". This assumes that energy levels are "occupied" by a limited number of actions. Why that would be true is entirely unclear to me.

For example, for any action with a certain energy level, there is a mirror action (with all joint torques multiplied by -1) with exactly the same energy consumption. Even if for some reason such specific pair of distinct actions with exactly the same effort
is impossible in a given setting, given how high the degree of freedom of the system is, it is still easy to imagine that every energy level offers a plethora of actions to choose from.

This raises a question of why then we see good performance in 2-action scenario. There is an alternative explanation that was not explored: there is a unique action with zero effort: doing nothing.
I suspect that it may be the only special case, where energy level is mapped uniquely onto an action (i.e. there is no different action with the same (0) energy penalty).

Another alternative explanation that must be addressed is the exposure bias. Rare intents are sampled less frequently, therefore the model has less time to optimize its actions for that intent. This may explain some of the differences in energy costs between the rare and frequent actions.

Lastly, the architecture choice and problem setting formalization also raise certain concerns. Specifically, the sender only receives joint positions and angles at time t, with no history. This does not seem to be an appropriate state specification. Coupled with the fact that the network is a simple feedforward architecture (no recurrence), this
limits the number of possible jestures that can be generated.

*** Conclusion ***

The paper addresses a highly relevant problem and proposes a number of interesting ideas. Unfortunately, in my opinion,
some of the proposed ideas (studying communication in an embodied setting, Zipfian intent distribution, costly actions), when introduced together, are not interacting well and are hindering result interpretation.

As authors mention, using an embodied setting with multidimentional continuous action and observation spaces makes the problem extremely challenging. Therefore, when we see only partial success or no success in learning, it becomes difficult to identify the true source of the problem. That is, it is unclear if we are hitting a fundamental limit of what can be achieved by the combination of the Zipfian intent + costly actions, or is whether it a just a limitation of the architectures/algorithms that the authors used, when those algorithms are applied in a challenging multidimentional setting.

The latter concern is exacerbated by some technical questions. In particular, the choice of a feedforward architecture seems extremely limiting, when combined with the state description that the authors used. The state space for the policy (sender) agent only includes the current joint configuration, without the history. This renders a large number of actions impossible (i.e. the "pendulum-like" wave of a hand becomes impossible, since when the hand returns to the middle after the first half-swing, the policy would act as if the action just started).

Lastly, the logic behind the main hypothesis is not fully theoretically supported, and the alternative explanations are not addressed.

Overall, I believe that the contribution is extremely promising, but it still requires some polishing and potential restructuring in order to be up to the standards of the ICLR conference. I do deeply hope to see this paper improved and published in the future, as I believe that after some improvements, it will be of interest to a large portion of the community.

*** Suggestions ***

I believe that the paper can be made much stronger if the authors take a step back and start with a simplified communication game setting. E.g. instead of actually producing an action trajectory, agents could pick among a number of actions with pre-defined effort levels, or they can separately pick the gesture and the amount of effort to allocate to that gesture (i.e. a subtle hand-wave vs flailing one's arms in the air).
This would allow to explore the benefits and limitations of the proposed Zipfian intent+Costly actions approach. In that case, demonstrating that the approach also works in an emboddied setting would be a beautiful cherry on top. Alternatively, it is possible to "boost" the embodiment part of the paper,
e.g. by pre-training the agents on some other tasks, thus making certain actions more natural, etc. Currently, it just seems that embodiment itself is not adding additional insights, but complicating the experiments.

Adding noize (mentioned in the appendix) is a very good idea as intuitively, it seems that it is the only reason why the agents don't converge on an array of extremely subtle actions with near-zero effort. I believe that it was not discussed enough in the paper and maybe its importance is somewhat overlooked.

*** Questions ***

- "Furthermore, with an increase in task complexity to 10 concepts, the external observer never performs better than this baseline strategy" I thought the baseline accuracy was 0.34, so it seems that with curriculum the external observer does outperform the baseline.
- Apart from the zero-effort action, why can't the agents converge on minimally distinguishable actions with the same effort level?

*** Typos and other minor suggestions ***

"execution decentralized" -> "decentralized execution"
"Agent reward" verbally described as a "function of state and actions taken", but the formula is written as if that it is a function of observation and actions taken.

Table 1 is slightly confusing to read, it may be better to make it broader so that the phrase "Train Input" does not break into two lines.

Conflicting notation: N is used as the number of agents on page 3, but later N refers to the number of concepts. 10 agents - not enough to infer the latent variable, especially if no randomness is added to the agents.

I personally believe that the use of the term "zero-shot" in this setting is not ideal. The observer is trained on a sample of communication protocols and is then tested on a held-out set.
In my view, it's analogous to "generalization performance", not "zero-shot" performance. I understand that the authors are referring to another study that did use "zero-shot", so this consideration did not affect my score.
Before the name for this problem setting is thoroughly established, it may be possible to use "out of sample coordination" or some other option.

*** Update after rebuttal ***

I have read the rebuttal and I deeply appreciate the detailed response by the authors. I especially appreciate the introduction of a number of experiments on a simple domain that help to illustrate the main point of the paper.

At the same time, I believe that some serious issues still remain unresolved. For example, my main concern remains: I believe that wrapping the problem in the embodied setting is not introducing additional insights. I must clarify that I fully agree that studying how embodiment affects cognition/behavior is an extremely important and exciting area of research. But in the present paper, the models have no chance to benefit from embodiment (since there is no prior / shared embodied experience), but rather have to solve the task despite being placed in an embodied setting.

The main insight (in my opinion) is the observation that Zipfian intent distribution together with energy costs could be a good basis for zero-shot communication. I think that additional experiments that the authors introduced help to strengthen this point, although more experiments could still be beneficial (e.g. systematically varying the population size), as well as a more thorough theoretical discussion. At the same time, the limitations of the main "embodied" experiment remain (most importantly, the fact that fairly high accuracy can be achieved because of the unique "do nothing" action trajectory).

In short, a large part of the paper (on embodiment) contributes relatively little in terms of its impact and conclusions that can be drawn from it, which necessarily limits the extent to which the main insightful point can be explored. The main point is truly interesting, however, which makes the paper borderline.

Overall, I believe that the paper is extremely promising and I would love to see an expanded version published. I feel very torn about the decision, but at the moment, I believe that the paper is still below the acceptance threshold, although only marginally. I am happy to adjust the score up, and I regret that I can not switch it to an "accept" recommendation.

As a minor aside - the newly introduced Colab Notebook does not fully run and crashes at the cell #4 (model loading), so I can not fully explore the newly introduced experiments. That being said, I think that after fixing, this resource can be very useful in the future. This minor issue did not affect my evaluation.

---

> ### Author Response · Authors · 2020-11-24
> **REBUTTAL (Reviewer 2): Part 2/2**
>
> Again thank you for your feedback. Our comments continued below.
>
> @“Providing effort information … defeating the purpose”: We understand your concern about providing effort information, particularly as it pertains to realistic downstream applications.  However, we want to emphasize that this work aims to answer more fundamental research questions about enabling general embodied communication protocols through continuous channels.  Self-play, combined with a high-dimensional continuous channel, yields a highly ill-posed optimization problem (many, many plausible protocols). And it is important to keep in mind that learning a communication protocol in a continuous action space is still an open question.  Thus, part of the fundamental research challenge here is in understanding how to guide the policy search within this high dimensional space toward protocols that can generalize to novel agents.
>
> With that, our analysis focuses on elucidating reasons for not seeing better zero-shot coordination when given trajectory input.  And decoupling challenges in optimization during protocol learning from challenges with inferring the latent structure were key to understanding how to enable better communication generalization with continuous-action protocols.
>
> @“Torque curriculum makes the results very specific to the considered setting”:  This is a valid concern.  To provide additional clarity and perspective though, the torque curriculum was used to understand if pretraining on one of the learning objectives (energy cost) could aid the policy parameter search by the optimization algorithm.  Its effectiveness (or lack thereof) then was intended to provide insights for the community towards developing methods that can automatically discover the maximum possible continuous-action coordination protocols.  However notably, with our updated results, this curriculum is no longer required for observing at least some communication generalization (on all tasks).
>
> @“...agent population considered in these experiments is 10”:  Yes, your intuition is correct that training these embodied protocols is highly time consuming and becomes quite computationally expensive.  Indeed, even when training on the order of hundreds of agents.  Notably, when we began this project, our experimental investigations for embodied protocol training used population sizes of 100 - 500 agents. However, protocol training time alone was on the order of several days longer, yet the mean trends we observed were relatively consistent (with 10, 100, and 500 agent populations).  So we decided to move forward with smaller population sizes in order to iterate more quickly and focus on making progress towards deeper analysis of the optimization challenges arising.  It’s of course possible that if we were to train on the order of thousands of agents in the population, we may observe qualitatively better generalization.  In fact, we have now included simple toy settings that train quickly. With those, we should be able to train 100s or 1000s of agents.  However, having to train population sizes of this magnitude using embodied channels (which require a kinematics module for generating motion) may render the use of emergent protocols computationally infeasible for many downstream applications of embodied agents (e.g. real robots).  A key insight from this work is that we need more robust methods for guiding the policy search towards globally optimal communication protocols in ZS coordination problem settings.
>
> @Suggestions about pretraining the agents on some other task to make trajectories more natural:  Great and insightful suggestion.  This has indeed come up in several conversations about future investigations. However, it is important to note that, while interesting to explore, a human-interpretable or “natural” protocol is not necessary for ZS agent-agent communication.  The two are orthogonal problems.  And as embodied protocol learning represents a highly underexplored problem setting in the space of multi-agent communication, learning self-compatible protocols is an important first step.

---

> ### Author Response · Authors · 2020-11-24
> **REBUTTAL (Reviewer 2): Part 1/2**
>
> We would like to thank you for your positive as well as all of your constructive feedback and suggestions.
>
> @Embodied setting: As expressed in the paper, we believe that embodied communication is both an important problem to study in AI and also naturally lends itself to the study of zero-shot communication based on common-knowledge costs, since these are derived directly from the physical properties of the setting.
>
> However, we entirely agree with the assessment that the paper currently addresses a combination of two difficult problems, which makes it more difficult to understand the relative challenges. To address this, as suggested by reviewer #2, we have now added a discrete toy-setting (in Sections 5.1.1, A.4, and A.5) that entirely focuses on zero-shot communication based on a non-uniform intent distribution and energy costs, without the confounding challenges of high dimensional optimization. This toy setting, which reproduces the main findings of the paper, is implemented in a Colab notebook and can be run quickly online, which hopefully will provide a great starting point for future investigation for other researchers.
>
> @”Many trajectories with the same energy”: This is entirely true: Indeed, in general there will be many trajectories that have the same energy and the only exception from this is the unique 0 energy trajectory. So there is no reason that independently trained optimal self-play policies should be able to coordinate, as was correctly pointed out by the reviewer.
>
> However, the problem setting in the paper is not the self-play setting but zero-shot coordination. We have updated the paper to emphasize and clarify this point. And indeed, it is clear that the optimal zero-shot policies are those that learn to ignore the spurious degrees of freedom in the trajectory and instead uniquely encode and decode intents based on energy. As we demonstrate in the paper, learning these optimal Zero-shot policies is an extremely difficult challenge.
>
> @Role of the noise: This is a great point! Indeed, the noise is absolutely crucial. Just like the uncertainty principle is required for making the information content in the universe finite, our finite noise level provides a ‘coarse graining’ of the distinguishable trajectories and thereby prevents a collapse of the trajectories. We have now clarified this point in the final paragraph of Section 3. Thanks again for the suggestion.
>
> @”zero-shot performance”: You make a great point. Indeed, the terminology in the first version of our paper was a little bit confusing in this regard. We have now cleaned up the terminology, hopefully this will address your concern: Zero-Shot Communication: This is our problem setting, a specific instance of Zero-Shot Coordination. The goal is to develop learning algorithms that allow independently trained agents to communicate by exploiting the structure in a common knowledge problem setting. Cross-Play performance: This is the evaluation metric for the ZSC setting. In cross-play we evaluate independently trained agents in novel constellations after training has finished.
>
> Note that these naming conventions are now consistent with the original paper that introduced Zero-Shot coordination. While we agree that ‘generalization performance’ is another possible term, we believe ‘cross-play’ is more specific and that it is good to keep the notation consistent with prior work to avoid confusion.
>
> @Feedforward model Architecture: This is another great point! In general, clearly recurrent policies will have a lot more degrees of freedom than feedforward policies. However, as pointed out above, in our problem setting the agents already have astronomically more degrees of freedom than required for the optimal policies. In fact, one of the great challenges is how to train policies that learn to ignore almost all of these degrees of freedom, discovering the fact that coordinating based on the energy values is the optimal strategy. Given that this task is currently beyond reach for feedforward policies, we believe it is prudent to leave the harder task (recurrent policies) for future work.
>
> @“Figure 4 is deeply confusing”: We understand how the snapshots may have caused confusion. We have now supplemented the figure caption with explanation about upsampling during visualization.

---

### Official Review · AnonReviewer4 · 2020-10-28
**Interesting field of research**

**Rating:** 6
**Confidence:** 1

**Review:**

##########################################################################

Summary:

The paper deals with agents that communicate non-verbally via actuating their joints in a 3D environment. The authors show that the agents should be able to learn protocols that can generalize to novel partners. Furthermore, the authors find that the current training approaches are brittle, and they propose and evaluate approaches to address this challenge.

##########################################################################

Reasons:

Overall, I vote for accepting. The authors develop and evaluate an algorithm that allows for emergent communication which generalizes to novel partners. Furthermore, the authors openly communicate and address the brittleness of the training.

##########################################################################

Pros:

* Novel algorithm to perform emergent communication via joint actuations
* Algorithm allows for generalization to novel partners
* Suggestions to improve training stability

##########################################################################

Cons:

* It would be desirable to train the agents N times with different initializations and report mean and std of the performance
* More experiments and evaluation results with various joint numbers, intents, potentially different communication strategies would be helpful

##########################################################################

Misc:

The text in Figure 1 is too small.

---

> ### Author Response · Authors · 2020-11-24
> **REBUTTAL (Reviewer 4)**
>
> We would like to thank you for your positive and constructive feedback..
>
> @“desirable to train the agents N times with different initializations…”:  We want to clarify the agents actually only differ in their model parameter initializations.  So in our experiments, having N training runs with k agents each is equivalent to initializing a population of N*k agents.
>
> @ “More experiments and evaluation results…”:  Thank you for the suggestion.  We agree that more experiments, particularly in a simpler discrete domain where different communication strategies could be pursued, is instructive and insightful.  We have now added new toy-setting experiments in Section 5.1.1 and Appendix Sections A.4, and A.5.
>
> @"text in Figure 1 is too small": We increased the size of this diagram, per your suggestion.  Hopefully it is more legible now.

---

### Official Review · AnonReviewer1 · 2020-10-30
**Zero-shot emergent non-verbal communication**

**Rating:** 6
**Confidence:** 3

**Review:**

The authors study the zero-shot emergent non-verbal communication in this paper. Different from most papers on emergent communication. this paper uses the motion of three-joint agents. The agents meet partners that they have never seen in the training phase, presenting the challenge of the universal protocol. To make a universal protocol possible, the authors study intents sampled from Zipf distribution and energy regulation. The authors conducted experiments on tasks with 2, 5, 10 intents. The results show that providing latent energy feature is essential for zero-shot coordination. To achieve better than chance accuracy on tasks with 10 intents, a torque curriculum is needed.

Pros:
1. The setting of ZS coordination with non-verbal emergent communication is novel and interesting.
2. The use of energy regulation and Zipf distribution is natural and intuitive. The results also support the intuition.

Cons:
What is the aim of using motion as a communication protocol? In human communication, actions are often used as an iconic gesture to describe intents that are not well described by words. In this sense, humans can perform zero-shot non-verbal emergent communication since the iconic gestures are always universal. In other words, a grounded world should be used to study ZS coordination. This paper seems to study a problem that is not realistic.

---

> ### Author Response · Authors · 2020-11-24
> **REBUTTAL (Reviewer 1)**
>
> We would like to thank you for your positive and constructive feedback.
>
> @the aim of studying motion as a form of communication: Motion is an important and commonly-employed form of implicit communication for biological agents (both humans and animals).  Thus, as AI research aims to move towards human-level intelligence and artificial agents compatible with humans, intentional movement-based communication represents an important skill set for embodied agents to learn.  Indeed, for some agents, it is a primary form of communication; this would be the case for many types of robotic agents (e.g. robotic manipulators), which are not expected to use language.
>
> @“...a grounded world should be used to study ZS coordination”:  We understand your concern here.  It is important to note though that since embodied emergent communication is a relatively unexplored problem in the multi-agent communication literature, we begin investigations with the simplest instantiation of the problem setting.  Using a more complex simulated world would require elements such as a 3D simulator with a physics engine, 3D object models, an agent perception module, localization in the simulated world, etc.  These are certainly interesting and relevant aspects to explore for future investigations, as this subfield grows, but adds undesired complexity for an initial investigation into communication through high-dimensional continuous channels. To provide grounding in the environment, we employ a universal (common-knowledge) energy cost, derived from first principles of motion (physics).  This is physical energy (computed as torque) exerted by agent limbs in order to generate communication, grounded in the world since biological agents must also exert physical energy to generate gesture-based communication.

---

### Author Response · Authors · 2020-11-24
**NOTE about Updated Experiments  (for all reviewers)**


We have now included additional experiments in a toy discrete-channel domain, at the suggestion of Reviewer 2, in order to decouple the difficulty of learning general protocols in zero-shot coordination problem settings from the additional complexities engendered by the use of embodied, high-dimensional continuous channels.  New toy-setting experiments and results are introduced in Section 5.1.1 (+ Fig 3) and elaborated on in more depth in Appendix Sections A.4 and A.5 (+ Figs 7->12).

We also provide a Colab notebook for experimenting with ZS coordination in the toy discrete domain explored.
It is viewable at the URL --  http://shorturl.at/luHPX.

Additionally, after submitting the initial paper, we ran more continuous-domain experiments and identified a minor issue with the evaluation protocol used in the original experimental results.  We’ve now updated the paper (specifically, Tables 1 and 2) to be consistent with the updated experiments.  Most of the trends remain consistent.  Two key differences however are (1) we now observe some zero-shot coordination on the N=10 task, even without a torque curriculum and (2) results both with and without the torque curriculum seem comparable, in most cases.  Thus, the curriculum is no longer required, though still reported.

Overall, our key findings are consistent: Inducing the energy-based latent structure is important for enabling zero-shot coordination - across both discrete and continuous domains and across all communication tasks tested.  Yet, using high-dimensional continuous channels, agents are currently not able to automatically discover this latent structure.  Even when given the latent structure, they do not learn an optimal policy reliably.  Thus, interesting open challenges persist.

---

### Decision · Program_Chairs · 2021-01-07
**Final Decision**

**Decision:**

Reject

**Comment:**

This paper received borderline scores, R1, R3, R4 gave a score of 6 and recommended a borderline acceptance. R2 provided by far the most detailed review and recommended a score of 5 (i.e., borderline reject). After the rebuttal, R2 comments, "I believe that the paper is still below the acceptance threshold, although only marginally". Overall, I concur with R2. The reasons are detailed below:

The paper proposes a method for communication between two agents, wherein one agent actuates its joints to communicate intent. Intuitively, this resembles making a gesture. The paper considers the setting of a discrete number of intents. The sender agent is modeled as a neural network that takes as input the intent and outputs a trajectory of joints. The receiver observes a noisy version of the trajectory and outputs the intent. The parameters of the sender policy and receiver discrimination network are optimized to maximize classification accuracy. It is shown that if the intents are sampled from Zipf distribution and trajectories are penalized based on their energy, then a receiver agent initialized from scratch is better at inferring the intent from a pre-trained speaker agent, as opposed to when the distribution is uniform or when the energy regularization is not used (Figure 2).

Further, section 5.2 shows that when the listener is provided with the energy of the trajectory then it is better at recognizing the intent as opposed to being provided with the entire trajectory when a number of intents are small. With a larger number of intents (N=10), the performance is at chance accuracy.

The biggest challenge with the paper is that it is very poorly written. Large parts of the method and experimental setup are in the appendix (A.2 / A.3), which makes it hard to understand the paper. Section 4.2 is rather confusing because the ideas introduced are not used for training, but simply for evaluation. Further, the authors point out in the rebuttal that torque curriculum is not required, but it is still there in the paper and makes it more confusing. I recommend the authors to substantially rewrite the paper and focus on relevant parts instead of philosophical arguments. Lastly, I am confused by results in Table 2 -- the authors mention in the text that with 10 intents, intent identification is at chance (i.e. 34% accuracy), but the table shows 56% accuracy. A clarification would be helpful here.

The problem of communicating intents via gestures, when the agents are unaware of mapping from intents to gestures is an exciting area of research. From the perspective of emergent gestures, this paper has a novel contribution. However, the settings are toy and even in such a setup, the results are underwhelming. The assumptions that make the setup toy are: the listener agent knows about all joint locations of the sender (with some noise) and also has access to the energy exerted by the agent. Without access to energy, the performance is poor. In real-world scenarios, these are big assumptions. Furthermore, even when the energy is known For instance, even when the number of intent is small (i.e.,  N=10,) the performance is bad. The authors argue that is due to local minima in the optimization -- but that's exactly where the contribution could have been.

I will reiterate, that the authors claim their contribution is in using energy minimization + Zipf intent distribution as a mechanism for communicating intent -- which I agree to. However, as pointed out earlier, the paper is not well executed or written and therefore I recommend rejection.